# Distributed Inference and Fine-tuning of Large Language Models Over The Internet

**Alexander Borzunov**[*]
HSE Univesity, Yandex

**Max Ryabinin**
HSE Univesity, Yandex

**Artem Chumachenko**
Neiro.ai

**Dmitry Baranchuk**
Yandex

**Tim Dettmers**
University of Washington

**Younes Belkada**
Hugging Face

**Pavel Samygin**
Yandex School of Data Analysis

**Colin Raffel**
Hugging Face

## Abstract

Large language models (LLMs) are useful in many NLP tasks and become more capable with size, with the best open-source models having over 50 billion parameters. However, using these 50B+ models requires high-end hardware, making them inaccessible to most researchers. In this work, we investigate methods for cost-efficient inference and fine-tuning of LLMs, comparing local and distributed strategies. We observe that a *large enough* model (50B+) can run efficiently even on geodistributed devices in a consumer-grade network. This could allow running LLM efficiently by pooling together idle compute resources of multiple research groups and volunteers. We address two open problems: (1) how to perform inference and fine-tuning reliably if any device can disconnect abruptly and (2) how to partition LLMs between devices with uneven hardware, joining and leaving at will. In order to do that, we develop special fault-tolerant inference algorithms and load-balancing protocols that automatically assign devices to maximize the total system throughput. We showcase these algorithms in PETALS[1] — a decentralized system that runs Llama 2 (70B) and BLOOM (176B) over the Internet up to $10\times$ faster than offloading for interactive generation. We evaluate the performance of our system in simulated conditions and a real-world setup spanning two continents.

## 1 Introduction

In recent years, the NLP community has found that pretrained language models greatly accelerated progress on many research problems through either fine-tuning (Radford et al., 2018) or simple prompting (Brown et al., 2020). Their quality tends to improve as we increase model scale (Radford et al., 2019; Kaplan et al., 2020). Following this trend, modern language models often have hundreds of billions of parameters (Brown et al., 2020; Rae et al., 2021; Zeng et al., 2021; Kim et al., 2021).

Most recently, several research groups open-sourced their pretrained LLMs with over 50B parameters (Zhang et al., 2022; BigScience, 2022a; Touvron et al., 2023a,b). However, they are still difficult to use due to the sheer size in terms of parameters. For example, OPT-175B and BLOOM-176B need over 350 GB accelerator memory for inference and even more for fine-tuning. As a result, even basic inference for these LLMs requires multiple high-end GPUs or multi-node clusters. Recent studies

---

[*]Correspondence to: `borzunov.alexander@gmail.com`

[1]PETALS source code and documentation are available at `https://petals.dev`

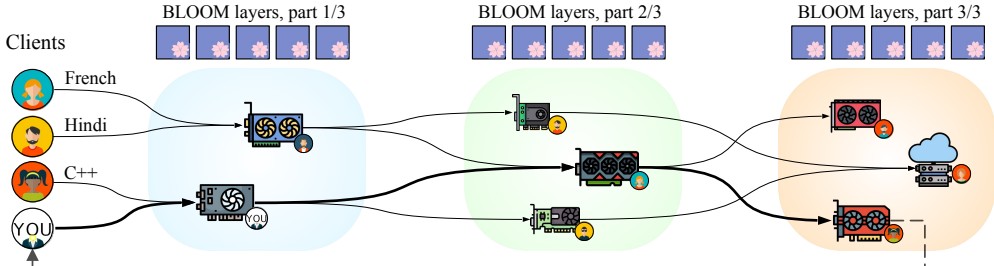

Figure 1: A high-level overview of our system design. Servers store pretrained LLM layers and temporarily hold attention caches for inferencing. Clients hold embedding layers and learned prompts/adapters (if used). Arrows denote temporary chains formed for inference.

propose algorithms for running large models with more affordable hardware (Pudipeddi et al., 2020; Ren et al., 2021), e.g. by offloading parameters to RAM. However, as we show in Section 3.1, these techniques are inefficient in many use cases, such as LLM-based chatbots and search engines.

In this work, we search for a more cost-effective way of running pretrained LLMs in their main use cases: inference, in-context learning, and fine-tuning. We analyze latency and throughput for these use cases and determine which factors become dominant for very large models. Notably, for models with over 50B parameters, communicating activations over a slow network can be faster than swapping layers from local RAM or SSD. Based on these observations, it should be possible to run LLMs cost-effectively by pooling together commodity hardware over the Internet.

However, existing LM algorithms are not designed to run inference with unreliable devices or high-latency networks. To bridge this gap, we formulate a novel algorithm for fault-tolerant distributed autoregressive inference of very large models. Using dual attention caches, this algorithm can quickly recover from a failed server and reassign the load to one or more replacement servers. Finally, to make sure that there are enough servers for every part of the model, we develop a decentralzied load-balancing algorithm that assigns transformer blocks to every server to maximize the total system throughput. The fully decentralized nature of these protocols allows participants to add or remove their devices at any point, making optimal use of GPU idle time.

We summarize the main contributions of this work as such:

- We analyze the problem of cost-efficient LLM inference and propose a novel algorithm that can inference large (50B+) language models on distributed unreliable devices. To the best of our knowledge, this is the first algorithm that can inference LLMs with 50B+ parameters in this setup.

- Using this algorithm, we develop PETALS — a decentralized system for inferencing and fine-tuning LLMs over the Internet. The system allows users to run inference and fine-tuning over a swarm of unreliable devices with the same correctness guarantees as when running locally. The system runs persistently with the help of volunteers.

- We benchmark the performance of the proposed algorithms on Llama 2 (70B) (Touvron et al., 2023b) and BLOOM (176B) (BigScience, 2022a). We run experiments in controlled conditions, with simulated network latency and server failures, and in the actual geo-distributed system spanning two continents. With realistic network speeds, our distributed algorithms perform autoregressive generation $\geq 10\times$ faster than local offloading.

## 2 Background: efficient training and inference

There is a wide variety of methods optimizing training and inference for most deep learning workloads. Here, we focus on two areas relevant for our analysis: model parallelism and parameter offloading.

### 2.1 Model parallelism

Model parallelism is a family of distributed training algorithms that assigns each device to hold a subset of model parameters, run a subset of computations and communicate output activations. *Tensor parallelism* assigns each device to compute a subset of each model layer (e.g., a subset of neurons), then communicate results between each other and proceed to the next layer (Krizhevsky et al., 2012;

Ben-Nun & Hoefler, 2019; Tang et al., 2020). Each device performs a symmetric computation, applied to a different slice of model weights, which makes tensor parallelism compatible with MPI-based communication. In turn, the main performance overhead of this strategy comes from all-to-all communication (and synchronization) after each layer (Krizhevsky, 2014).

*Pipeline parallelism* reduces the communication overhead by assigning each device with one or several full layers (Huang et al., 2019; Narayanan et al., 2019; Yang et al., 2019). During the forward pass, each stage applies its subset of layers to the inputs supplied by the previous stage, then sends the outputs of the last layer to the next stage. For the backward pass, this process is reversed, with each pipeline stage passing the gradients to the same device that previously supplied it with input activations. To better utilize the available devices, the pipeline must process multiple microbatches per step, allowing each stage to run in parallel on a different batch of inputs. Even with optimal execution, some of the pipeline stages will remain idle some of the time (Huang et al., 2019).

Both of these strategies are actively used for training LLMs. Real-world distributed training systems usually combine multiple forms of parallelism depending on hardware and network type (Narayanan et al., 2021; Rajbhandari et al., 2020; Jia et al., 2019). Tensor parallelism is typically used within a single multi-GPU server or closely interconnected TPU cores (Narayanan et al., 2021; Shazeer et al., 2018). In turn, pipeline parallelism is used to connect multiple servers (Narayanan et al., 2021). Recent works demonstrate that model parallelism can be used for cost-efficient *pre-training* of LLMs by pooling together idle GPU devices (Athlur et al., 2022; Wang et al., 2022; Kuszmaul, 2022; Yuan et al., 2022; Ryabinin et al., 2021).

## 2.2  Offloading

Parameter offloading relegates model parameters from accelerator memory to a slower but cheaper storage: typically RAM or SSD (Pudipeddi et al., 2020; Ren et al., 2021; Rajbhandari et al., 2021). When using the model, parameters are loaded to the accelerator just-in-time for computation, one or few layers at a time. In principle, this method allows running large models with a single low-end accelerator as long as there is enough RAM (or SSD) to store the model.

The main drawback of this strategy is having to load and unload through all model parameters for each forward and backward pass, which can be time-consuming. This extra time can be amortized in workloads where model can do a lot of useful computations for each time a parameter is loaded. In practice, using offloading to run a single token through the OPT-175B on one GPU in the best-case scenario of hardware and bandwidth[2] would require 11 seconds per forward pass, or twice that for training. As we show in Section 4, real-world performance is significantly slower.

Pudipeddi et al. (2020) circumvents this by training with very large batches, and hence, increasing the computation. In turn, Ren et al. (2021); Rajbhandari et al. (2021) reduce the overhead by overlapping communication and computation, that is, doing useful computation for the current layer while waiting for the transfer of the next layer to finish. Some of these systems Ren et al. (2021) also partition offloaded parameters between devices. However, unlike model-parallel training, distributed offloading still requires each device to compute the full model.

## 3  Method

Using pretrained large language models for NLP tasks consists of two main workloads: inference and fine-tuning. The inference workload typically consists of encoding an input text, then generating tokens autoregressively. In turn, fine-tuning requires updating either all of the model's parameters or (more commonly for large models) a small set of trainable weights (e.g., adapters or soft prompts) by backpropagation. These two workloads also cover more advanced use cases:

- Manually engineering prompts for a given task, then deploying the model with these prompts.

- Fine-tuning with adapters (Hu et al., 2021; Houlsby et al., 2019; Liu et al., 2022b) or "soft" prompts (Liu et al., 2021b; Lester et al., 2021; Liu et al., 2021a) and inferencing fine-tuned models.

- Distillation into a smaller task-specific model for faster inference (Schick & Schütze, 2021).

---

[2]Specifically, 16-bit parameters, PCIe gen. 4 at 31.5 GB/s (16 lanes), infinite compute and memory bandwidth.

Counter-intuitively, we found that inference is more challenging than fine-tuning for cost-efficient setups. To that end, we dedicate most of this section to inference-specific problems. As for fine-tuning, we describe a way to support arbitrary parameter-efficient fine-tuning in Section 3.4.

## 3.1 Performance bottlenecks of LLM inference

Unlike training, autoregressive LLM inference cannot be done with a single pass through the model. Instead, the model needs to process one token at a time, pass it through the entire model, then generate the next token and repeat the process. In case of model parallelism, training an $n$-layer[3] model on a sequence of $t$ tokens needs $O(n)$ communication rounds, while generating the same sequence needs $O(n \cdot t)$ rounds, making it more susceptible to network latency. Similarly with parameter offloading, generating a sequence of $t$ tokens needs loading every layer $t$ times, which also takes $O(n \cdot t)$ time.

The other problem of autoregressive generation is dealing with attention for past tokens (Vaswani et al., 2017). During an inference step $t$, each layer needs to attend to $t - 1$ previous attention keys and values. Existing inference algorithms store past entries in accelerator memory. Caching half-precision activations of a 2048-token sequence for large models like GPT-3 (Brown et al., 2020) or OPT-175B (Zhang et al., 2022) (with 96 layers of 12288 units each) takes up 9.6 GB GPU memory *for each sequence*. Offloading these cached values faces the same problems as offloading in general.

An alternative solution is to recompute all previous tokens on every inference step, storing only one set of keys & values at a time. Naturally, this approach needs increasingly more computation with sequence length $t$, for a total of $O(t^3)$ time for transformer-based models[4]. Surprisingly, this approach is often more efficient than offloaded caching, especially for shorter sequences due to the overhead from loading and storing cache from RAM or SSD.

Parameter offloading can still be efficient when generating *large amounts of short sequences* in bulk. Each individual sequence still takes a long time to generate, but the system maintains high throughput by running many samples in parallel. Unfortunately, this scenario does not cover many important LLM use cases. For instance, it is incompatible with in-context learning or prompt engineering, where the model needs to process long sequences of training examples (Brown et al., 2020). More importantly, it does not support "interactive" applications where LLM needs to quickly respond to a user input. This rules out many LLM applications such as conversation systems or input completion (e.g. ChatGPT or Smart Compose).

Hence, we explore a new solution based on pipeline-parallelism. A related line of work (Aminabadi et al., 2022) investigates model parallelism to inference LLMs in GPU clusters. However, their approach does not apply to our more affordable setups: cheap "preemptible" instances or connecting existing resources over the Internet. To operate in these conditions, an inference algorithm needs to deal with node preemption, network errors, and high latency.

## 3.2 Distributed generation with fault tolerance

In this section, we formulate an algorithm for inferencing LLMs in a fleet of unreliable geographically distributed devices connected over the Internet. Each device can act as a server, a client, or both. A **client** is a node operated by the user, which runs inference or fine-tuning jobs through the swarm of servers. A client only holds input and output embeddings ($< 3\%$ of model weights for BLOOM-176B) and delegates running transformer blocks (the most expensive computations) to remote servers. A **server** is a GPU-enabled node holding a set of consecutive transformer blocks and processing requests coming from client nodes.

For simplicity, we assume that every block is hosted on several servers and examine this assumption in the next section. Following this notation, a fault-tolerant algorithm should allow each client to complete an inference job with reproducible results even if some remote servers fail during inference.

As we discuss in Section 3.1, autoregressive generation requires many sequential communication rounds, making it sensitive to network latency. However, if every device stores its past attention

---

[3]Here and below, the term *model layer* (or *block*) refers to one transformer block that typically combines self-attention, a feed-forward network, normalization layers, and a residual connection (Vaswani et al., 2017).

[4]All public LLMs with 100B+ parameters use standard attention that scales as $O(n^2)$ for sequence length $n$.

cache, every round only transfers activations for a single token, i.e. several kilobytes of data[5]. We use this model to directly minimize the inference time over possible pipeline configurations. As we show later in Section 4.2, this allows efficient inference over a low-bandwidth Internet connection.

A more challenging problem is how to recover from node and network failures. If a remote server shuts down, any cached attention keys stored on that server will be lost with it. There are two naïve solutions to this problem: restarting inference from scratch or recomputing past embeddings on every step. Restarting might be enough at a small scale. However, running 50B+ models may involve many unreliable devices, making it unlikely to generate long sequence without at least one failure. In turn recomputing past attention caches requires communicating past tokens on every communication round, resulting in $O(n \cdot t^2)$ total data transferred, where $n$ is the number of pipeline layers and $t$ is the sequence length. In other words, both these solutions struggle to generate long sequences.

We address this problem by maintaining two types of cache: *server-side cache* holds past attention keys and values for their layers, like in existing inference algorithms, while *client-side cache* holds past inputs sent to a given pipeline stage[6]. If a server disconnects, a client can find another server with that pipeline stage and use client-side cache to restore the server state.

The resulting procedure is described in Algorithm 1. For every pipeline stage, the client maintains a heap (priority queue) of servers that hold this stage (and may hold additional stages). The servers in queue are ordered by the network latency, measured from past communication. These queues are maintained through the lifetime of a client. To begin generation, the client runs a beam-search-like procedure to find a sequence of servers that results in the least total inference time under our performance model. When running inference steps, a client keeps track of intermediate activations sent between pipeline stages. If a remote server fails or leaves, the client retrieves the next best server (or multiple servers) and requests it to restore the attention state from the client's cached activations.

When servers fail, the algorithm needs to send $O(t)$ data (in one round) for each failed server and compute only the stages held by the failed servers. This can be seen as an interpolation between naive and cached inference, depending on the server failure rate. If none of the servers fail, we recover $O(n \cdot t)$ communication, similarly to Aminabadi et al. (2022). In turn, if all servers fail after one step, the algorithm effectively performs non-caching generation, which is the best option in that scenario.

In the basic formulation, all communication between pipeline stages is routed through the client, i.e. the client receives the outputs of every pipeline stage, caches it and sends it to the subsequent stage. In practice, it is more efficient to let pipeline stages communicate directly: once the server obtains output activations, it sends them to both client and the subsequent stage. This reduces the total step time since both messages are a few kilobytes in size an can be sent in parallel. To verify that both client and the next pipeline stage received the same set of activations, they can verify the checksums (i.e. hash values) of the received activations asynchronously, without blocking computation.

Algorithm 1 can support greedy inference or any sampling variants (including Holtzman et al. (2020)). However, it requires one more step to support search-based algorithms such as beam search: cache reordering. This allows a client to generate multiple continuations of the same input prefix by cloning its attention cache and dropping less likely hypotheses. We describe beam search in Appendix C.

**Shortest path routing.** In the Algorithm 1, the `find_best_chain` function (line 4) selects a sequence of servers that can run the required layers in the least amount of time. To estimate this time we add up two factors: computation time, determined by server's compute throughput ("GPU speed") and the network latency between the client and that server. Servers measure their own compute throughput and share this information with the clients. In turn, clients measure the network latency between them and a given server by "pinging" the candidate servers during routing. If a server runs multiple consecutive blocks, we multiply the computation time by the number of blocks.

To find the best chain of servers, clients find the shortest path between the first and last block, using a graph where edge weights correspond to server inference time, as described in the previous paragraph. To minimize overhead, we do not run pathfinding from scratch on each call to `find_best_chain`. Instead, clients run lifelong pathfinding in the background and reuse it between inference calls. More specifically, we use the D* Lite (Koenig & Likhachev, 2005) algorithm because it allows clients to quickly adjust paths after a server is banned or leaves the network.

---

[5]For GPT-3 and OPT-175B, one 12288-dimensional token embedding in 16-bit precision takes up 24 KiB.
[6]Here, a *pipeline stage* is a set of consecutive model layers hosted on one server (as in pipeline parallelism).

**Algorithm 1** Generating sequence, client-side code

**Input:** prefix_tokens, embeddings, known_servers
```
 1: generated_sequence = list()
 2: cache = dictionary()
 3: streams = dictionary()
 4: chain = find_best_chain(known_servers)
 5: for server ∈ chain do
 6:     streams[server] = rpc_inference(server)
 7:     cache[server] = list()
 8: end for
 9:
10: inputs = embeddings(prefix_tokens)
11: while should_continue(generated_sequence) do
12:     tail_servers = copy(chain)
13:     while not empty(tail_servers) do
14:         server = tail_servers.pop_left()
15:         try:
16:             ▷ Attempt normal inference
17:             outputs = streams[server].send(inputs)
18:             cache[server].append(inputs)
19:             inputs = outputs
20:         catch ServerFailed:
21:             ▷ Replace the failed server
22:             streams.pop(server).close()
23:             past_inputs = cache.pop(server)
24:             new_servers = replace_failed_server(
25:                 server, past_inputs, cache,
26:                 streams, known_servers)
27:             chain.replace(server, new_servers)
28:             tail_servers.push_left(new_servers)
29:     end while
30:
31:     logits = compute_logits(outputs, embeddings)
32:     next_token = choose_next(logits) {e.g. greedy}
33:     generated_sequence.append(next_token)
34:     inputs = embeddings(next_token)
35: end while
36:
37: for server ∈ chain do
38:     streams[server].close()
39: end for
40: return generated_sequence
```

**Algorithm 2** rpc_inference(server)

**Input:** local_layers, stream
```
 1: cache = dictionary()
 2: for layer ∈ local_layers do
 3:     cache[layer] = make_empty()
 4: end for
 5: while not stream.closed() do
 6:     inputs = stream.receive()
 7:     for layer ∈ local_layers do
 8:         past_kv = cache[layer]
 9:         inputs, new_kv = forward(
10:             layer, inputs, past_kv)
11:         cache[layer].append(new_kv)
12:     end for
13:     stream.send(inputs)
14: end while
```

**Algorithm 3** replace_failed_server(...)

**Input:** server, inputs, cache, streams, known_servers
```
 1: known_servers.ban(server)
 2: missing_layers = get_layers(server)
 3: chains = select_by_layer(
 4:     known_servers, missing_layers)
 5: chain = find_best_chain(chains)
 6: replacements = list()
 7: while not empty(chain) do
 8:     s = chain.pop_left()
 9:     try:
10:         streams[s] = rpc_inference(s)
11:         outputs = streams[s].send(inputs)
12:         replacements.append(s)
13:         cache[s] = inputs
14:         missing_layers.pop(get_layers(s))
15:         inputs = outputs
16:     catch FailedRPC:
17:         known_servers.ban(s)
18:         chains = select_by_layer(
19:             chains, missing_layers)
20:         chain = find_best_chain(chains)
21: end while
22: return chain
```

## 3.3 Automatic load balancing

In order to run inference or fine-tuning, each server needs to be assigned to a pipeline stage, then reassigned if other servers join or leave the network. For example, if we deploy an LLM on idle compute resources from several data centers or labs, the number of participants may change over time based on the demand. Moreover, servers may have different compute throughput, network bandwidth, and geographical location. To operate in these conditions efficiently, servers should automatically choose which model layers they should serve in a given situation.

To that end, servers periodically run a load balancing procedure and switch to new blocks if necessary. Formally, servers choose blocks so as to maximize the total system throughput (tokens per second). Each server periodically announces its blocks and empirically measured throughput to a distributed hash table (Maymounkov & Mazieres, 2002). When a new server joins, it uses this information to identify a contiguous interval[7] of blocks that would increase the total system throughput the most.

---

[7]This interval is always contiguous, since splitting it would harm the inference latency.

Since peers may leave or fail at any time, all nodes periodically check if launching a rebalancing procedure would significantly improve the overall throughput. If it is the case, they switch layers until the throughput becomes near-optimal. In particular, if all peers serving certain blocks suddenly leave the system, this procedure quickly redistributes the remaining resources to close the emerged gaps.

We provide a detailed description of the load balancing algorithms in Appendix D and validate their properties in experiments reported in Appendix E.

## 3.4 Parameter-efficient fine-tuning

While LLMs achieve high quality on many problems with simple prompt engineering (Brown et al., 2020), they often need training to achieve the best results. Traditionally, this is done by fine-tuning all model parameters on the downstream task. However, for extremely large models, this strategy becomes impractical due to hardware requirements. For example, fine-tuning BLOOM-176B with Adam would require almost 3 TB of GPU memory to store the model, gradients, and optimizer states.

Fortunately, *parameter-efficient fine-tuning* methods have been developed that keep most of the pretrained model intact. Some of them choose a subset of existing parameters to update (Sung et al., 2021; Guo et al., 2021) while others augment the model with additional trainable weights (Hu et al., 2021; Houlsby et al., 2019; Liu et al., 2021b; Lester et al., 2021; Liu et al., 2021a, 2022b). Despite their lower memory requirements, parameter-efficient approaches are often competitive with full model fine-tuning (Hu et al., 2021; Liu et al., 2021a; Yong & Nikoulina, 2022) and even outperform it in low-data regimes (Liu et al., 2022a). Another appealing property of these approaches for our use-case is that they allow rapidly switching a pretrained LLM between adapters.

By focusing on parameter-efficient fine-tuning, we are able to simplify the system design by *making clients responsible for storing their trainable parameters* (see Figure 1). Servers can run backpropagation through their layers and return gradients with respect to activations, but they *do not update the server-side parameters*. Even when client communicates learned values (e.g. soft prompts) to a server, the server treats these values same as input activations. Thus, a server can simultaneously run different fine-tuning tasks without them interfering with one another. This design choice also allows users to define custom adapters in simple PyTorch without having network engineering expertise.

Unlike inference, fine-tuning forward and backward passes process the entire batch at one go and do not need to store past attention caches between successive client requests. Thus, in case of a failure, we can discard the incomplete forward/backward pass and just repeat the previous forward/backward pass request. This algorithm behaves similarly to the cache-less baseline from Section 4.1.

## 3.5 Implementation details

Since our main intended use-case is running on inexpensive low-end devices, we need to work around their capabilities. In terms of raw FLOPs, even consumer-grade GPUs like GeForce RTX 3070 could run a complete inference step of BLOOM-176B in less than a second (NVIDIA, 2020). However, the GPU memory can only hold a small fraction of model layers: running naïvely would require 44 RTX 3070 GPUs and 44 communication rounds. To make this more efficient, we use quantization to store more parameters per GPU, reducing the number of consecutive devices and communication rounds.

One option for quantization is to use 8-bit mixed matrix decomposition for matrix multiplication to quantize the weights to 8-bit precision and reduce the memory footprint compared to 16-bit weights, as suggested in Dettmers et al. (2022a). This decomposition separates hidden states and weights into two portions: about 0.1% of 16-bit outlier and 99.9% of 8-bit regular values, which roughly halves the memory footprint with negligible effect on the model quality (see evaluations in Appendix A). Another option is to use the 4-bit NormalFloat format (Dettmers et al., 2023).

To send less data between subsequent pipeline stages, we apply dynamic blockwise quantization (Dettmers et al., 2022b) to the hidden states before pipeline-parallel communication, which halves the bandwidth requirements without any noticeable effect on generation quality (Ryabinin et al., 2021). During fine-tuning, we also take advantage of gradient checkpointing (Griewank & Walther, 2000; Chen et al., 2016) and half precision to reduce VRAM usage — both are standard practice for large language models (Narayanan et al., 2021; Brown et al., 2020; Athlur et al., 2022). In experiments, we apply the same optimizations to baseline systems for a fair comparison.

Table 1: Sequential inference speed (steps/second) of BLOOM (7.1B) with varying failure rates. A failure rate $p$ means that sending any set of activations to the next stage of the pipeline fails with probability $p$. Missing values mean that the algorithm did not finish within 1 hour.

| Inference Algorithm | 128 tokens, failure rate: | | | | 1024 tokens, failure rate: | | | |
|---|---|---|---|---|---|---|---|---|
| | 0 | 1e-4 | 1e-3 | 1e-2 | 0 | 1e-4 | 1e-3 | 1e-2 |
| Caching with restarts | 17.1 | 16.7 | 12 | 0.18 | 15.5 | 11.8 | 0.48 | – |
| Cache-less inference | 3.44 | 3.44 | 3.44 | 3.44 | 0.89 | 0.89 | 0.89 | 0.89 |
| Algorithm 1 (ours) | 11.4 | 11.4 | 10.6 | 3.38 | 10.7 | 10.7 | 7.76 | 2.17 |

## 4 Experiments

### 4.1 Inference with unreliable servers

First, we conduct small-scale preliminary experiments to test the fault-tolerant generation algorithm described in Section 3.2. For these experiments, we use a smaller BLOOM model with 7.1 billion parameters (BigScience, 2022b). This model contains 30 transformer blocks with hidden size 4096. We compare our algorithm with baselines when generating a single sequence of length 512. For simplicity, we run all computations and communications in single precision and disregard word embeddings and logits for this set of experiments. We measure the time to run a certain number of tokens through all blocks and simulate failures by resetting pipeline stages at a certain rate.

We compare three inference strategies:

1. **Caching with restarts**, which refers to standard inference with servers storing attention caches. On failure, it restarts the entire generation from scratch since the failed server's caches are lost.
2. **Cache-less inference**, which reruns past tokens on every step. On failure, it restarts only the last generation step.
3. **Algorithm 1**, which is specifically designed for fault-tolerant inference.

All runs use four pipeline stages with (8, 7, 8, 7) model layers per pipeline stage. Each pipeline stage is served by a single GeForce 1080 Ti GPU; the four GPUs are running in a single system with dual Xeon Gold 6148 CPU, 12 DDR4 LRDIMM sticks with 64 GB each. The system has 16 dedicated PCIe Gen. 3 lanes per GPU in dual root configuration, without using PCIe switches. Each stage runs in an isolated Docker containers with virtual network interfaces, but there is no limit to communication bandwidth for this experiment. We repeat all experiments 50 times and report the average time. The adjusted standard deviation never exceeds 0.2%. We use the pipeline parallelism implementation from Megatron-DeepSpeed (BigScience et al., 2022) for the cache-less baseline.

We report performance measurements in Table 1. Unlike baselines, our algorithm provides reasonable performance *in all tested conditions*, especially for higher failure rates (common for communicating over the Internet, using spot/preemptible instances or unreliable hardware). Caching with restarts is most efficient for inference without failures, with our algorithm being somewhat slower due to less mature implementation. Finally, the cache-less inference can be competitive for short sequences (128 tokens), but slows down considerably on 1024 tokens, which agrees with our intuition from 3.1.

We provide plots showing additional evaluations for a wider range of failure rates (up to 5%) and sequence lengths (up to 2048 tokens) in Appendix F (Figure 3).

### 4.2 Experiments for Llama 2 (70B) and BLOOM (176B)

In this section, we evaluate our system on more practical tasks of running Llama 2 (70B) (Touvron et al., 2023b) and BLOOM (176B) (BigScience, 2022a). First, we consider servers running in a network with controlled bandwidth and latency[8]. We measure performance for **(a)** Llama 2 distributed across 3 servers with a T4 GPU each, **(b)** BLOOM distributed across 3 servers with an A100 (80 GB) GPU each, and **(c)** BLOOM distributed across 10 servers with an RTX 3090 GPU each. We use 4-bit NormalFloat quantization (Dettmers et al., 2023) for Llama 2 and 8-bit matrix decomposition (Dettmers et al., 2022a) for BLOOM in all evaluations including the baselines below.

---

[8]We simulate network conditions using `tc qdisc`.

Table 2: Performance of Llama 2 (70B) sequential inference steps and parallel forward passes. The network parameters refer to bidirectional bandwidth and round-trip latency (RTT).

| GPUs | Clients | Bandwidth | RTT | Sequential inference (steps/s, each client) | | Parallel forward (tokens/s, each client) | |
|---|---|---|---|---|---|---|---|
| | | | | Sequence length | | Batch size | |
| | | | | 128 | 2048 | 1×128 | 64×128 |
| 3× T4 (16 GB) | 1 | 1 Gbit/s | < 5 ms | 2.29 | 2.02 | 45.4 | 155.1 |
| | 1 | 100 Mbit/s | < 5 ms | 2.29 | 2.01 | 37.5 | 140.2 |
| | 1 | 100 Mbit/s | 100 ms | 1.57 | 1.44 | 23.7 | 128.7 |
| | 3 | 1 Gbit/s | < 5 ms | 2.02 | 1.74 | 21.2 | 124.2 |
| | – | Offloading | | 0.139 | 0.139 | 18.0 | 139.9 |

Table 3: Performance of BLOOM (176B) sequential inference steps and parallel forward passes.

| GPUs | Clients | Bandwidth | RTT | Sequential inference (steps/s, each client) | | Parallel forward (tokens/s, each client) | |
|---|---|---|---|---|---|---|---|
| | | | | Sequence length | | Batch size | |
| | | | | 128 | 2048 | 1×128 | 64×128 |
| 3× A100 (80 GB) | 1 | 1 Gbit/s | < 5 ms | 1.71 | 1.54 | 70.0 | 253.6 |
| | 1 | 100 Mbit/s | < 5 ms | 1.66 | 1.49 | 56.4 | 182.0 |
| | 1 | 100 Mbit/s | 100 ms | 1.23 | 1.11 | 19.7 | 112.2 |
| | 3 | 1 Gbit/s | < 5 ms | 1.65 | 1.49 | – | – |
| | – | Offloading | | 0.0495 | 0.0495 | 2.5 | 152.4 |
| | – | Local PP (NVLink) | | 2.46 | 2.28 | 98.4 | 279.5 |
| 10× RTX 3090 (24 GB) | 1 | 1 Gbit/s | < 5 ms | 1.65 | 1.54 | 59.1 | 230.1 |
| | 3 | 1 Gbit/s | < 5 ms | 1.65 | 1.54 | 54.7 | 221.4 |
| | 10 | 1 Gbit/s | < 5 ms | 1.17 | 1.01 | 31.0 | 131.0 |
| | 10 | 100 Mbit/s | < 5 ms | 1.05 | 0.99 | 20.1 | 28.1 |
| | 10 | 100 Mbit/s | 100 ms | 0.34 | 0.33 | 6.5 | 16.8 |
| | – | Offloading | | 0.0427 | 0.0427 | 2.2 | 109.3 |
| 12× heterogeneous (virtual servers) | 1 | 1 Gbit/s | < 5 ms | 1.24 | 1.06 | 37.9 | 180.0 |
| | 1 | 100 Mbit/s | < 5 ms | 1.24 | 1.05 | 25.6 | 66.0 |
| | 1 | 100 Mbit/s | 100 ms | 0.57 | 0.53 | 5.8 | 44.3 |
| | 12 | 1 Gbit/s | < 5 ms | 0.90 | 0.86 | – | – |
| 14× heterogeneous | 1 | Real-world setup | | 0.83 | 0.79 | 32.6 | 179.4 |
| Theoretical-best | – | Offloading | | 0.18 | 0.18 | 2.7 | 170.3 |

We report performance of:

- **Sequential (autoregressive) inference** for batch size 1 (i.e., each step generates 1 token). It is measured in generation steps per second a client can do and shows the *generation latency*.
- **Parallel forward passes** for batches of 128-token sequences[9]. It is measured in tokens per second a client can process. This shows the *system's throughput* during batch processing and fine-tuning.

Since the backward pass performance depends on a set of trainable weights, batch size, and other hyperparameters, we report its performance in different setups separately in Appendix G.

**Concurrent clients.** We also investigate the effect of having concurrent clients. We assume that each server belongs to a different person, and multiple people (possibly, all of them) are interested in running inference or fine-tuning at the same time. In order to do that, they run the client interacting with our distributed system. The client runs on the same machine, uses 8 CPU cores and no GPU. We report the speed of sequential inference and parallel forward passes that *each client gets on average*.

---

[9]Intenally, large batches are split into micro-batches of 1024 tokens each to minimize pipeline bubbles.

**Offloading baseline.**    We also evaluate parameter offloading, where each user runs independently on a single GPU, swapping parameters from CPU memory. First, we report the actual throughput of RAM offloading in case of DeepSpeed with default recommended parameters and enabled `pin_memory` (gives $1.2-2\times$ speedup). Next, we report the *theoretical-best* throughput the offloading baseline can reach for BLOOM. It is calculated as a maximal throughput in the best hardware setup possible (CPU RAM offloading via PCIe 4.0 with 16 PCIe lanes), assuming infinite GPU performance. The calculations are detailed in Appendix B.

**Local pipeline parallelism (NVLink).**    Next, we report performance for BLOOM running on a server with $3\times$ A100 (80 GB) GPUs. In this setup, a single server has enough GPU memory to load the entire model, which provides an *upper bound* for performance reachable with these GPUs. This setup runs pipeline-parallelism from DeepSpeed v0.7.7.

**Heterogeneous servers.**    To validate that our system works on heterogeneous hardware, we simulate 12 heterogeneous devices by partitioning each A100 (80 GB) into several virtual servers (3 large and 1 small). We get 9 servers hosting 7 blocks each, one server with 3 blocks and two more servers with 2 blocks (70 blocks in total, as required for BLOOM). Additionally, we benchmark the system on real heterogeneous GPUs with diverse compute capabilities in the "Real-world setup" below.

**Real-world setup.**    Finally, we benchmark BLOOM in a real-world setup with 14 smaller servers holding $2\times$RTX 3060, $4\times$2080Ti, $2\times$3090, $2\times$A4000, and $4\times$A5000 GPUs. These are personal servers and servers from university labs, spread across Europe and North America and connected to the Internet at speeds of 100–1000 Mbit/s. Four of the servers operate from behind firewalls[10].

**Analysis.**    We report the results for Llama 2 in Table 2 and for BLOOM in Table 3. For inference, performance does not depend much on bandwidth or sequence length but degrades with higher latency. In turn, fine-tuning forward passes for large batches are affected by both bandwidth and latency.

We can see that the offloading baseline is about an order of magnitude slower than our system for inference, both in practice and in the theoretical-best setup assuming an infinite GPU performance. For parallel forward passes, offloading is competitive if networking is limited to 100 Mbit/s or has high latency. In other cases, our algorithm offers higher throughput than offloading for training.

Crucially, our system significantly outperforms offloading even when each GPU node runs its own client doing single-batch inference at the same time. Thus, **given the same hardware**, a group of researchers will get much better inference speed by collaborating over the Internet using our system compared to each of them running offloading independently.

Finally, the real-world setup turns out to be slower than the A100 benchmarks due to slower hardware. Still, our algorithm outperforms offloading even when communicating between different continents.

**Additional experiments.**    We conduct two additional experiments to test individual components of our system. We evaluate the load balancing from 3.3 in isolation in Appendix E. We also evaluate the performance of model compression from Section 3.5 in Appendix A. To reiterate, for each model, we use the same compression strategy in our system and all baselines. Finally, we perform a qualitative evaluation of fault tolerance by shutting down random servers during inference and fine-tuning to verify that the algorithm produces correct outputs and gradients.

## 5   Conclusion

In this paper, we introduced a novel fault-tolerant algorithm for inferencing large language models. On top of it, we introduced a decentralized system for running LLMs on distributed unreliable devices connected over the Internet, which significantly outperforms other approaches to running inference on consumer-grade hardware. We demonstrated that the proposed system can scale to the largest publicly available language model with hundreds of billions of trainable parameters.

While our work is focused on technical aspects, it is important to consider limitations of our approach, such as privacy of data processed by outside peers, as well as broader impact of making LLMs more accessible. We discuss these issues and outline directions for future work in Appendix H.

---

[10]We use the Circuit Relay protocol from libp2p (libp2p, 2022) to traverse NATs and firewalls.

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

# Appendix

## A   Quality and efficiency of BLOOM with 8-bit quantization

As shown in Table 4, this method has little effect on LLM quality for major benchmarks. In terms of inference time, Table 5 demonstrates that quantization has about $5\%$ of overhead with batch size 1 (20 tokens), but becomes negligible for larger batches.

## B   Estimating theoretical best throughput with RAM offloading

In this estimate, we use the best possible hardware setup for offloading: CPU RAM offloading via PCIe 4.0 with 16 PCIe lanes per GPU. In 8-bit, the model uses 1 GB of memory per billion parameters, and PCIe 4.0 with 16 lanes has a throughput of 256 Gbit/s. We assume an offloading latency of zero in the upper bound estimation. As such, offloading 176B parameters takes at least:

$$\frac{176\,\text{GB} \cdot 8}{256\,\text{Gbit/s}} = 5.5 \text{ seconds}$$

This gives the upper bound of $1/5.5 \approx 0.18$ tokens/s for the inference speed.

## C   Extension to beam search algorithms

There are several variations of beam-search algorithm used for language model inference, including standard beam search, diverse beam search, constrained beam search, and more. A common thread between those algorithms is that they maintain a fixed number $k$ of candidate sequences between steps. These sequences are informally referred to as the "beam". On every step, these algorithms generate possible continuations of sequences in the previous beam, then use some fitness criterion to select $k$ of these continuations for the next beam.

From a computational point of view, this procedure is similar to simple "greedy" inference with a batch of $k$ sequences. However, there is one important difference: unlike batched inference, beam search algorithms can "shuffle" candidate sequences between steps. In other words, 3rd best sequence from time step $t$ can produce 1st or 2nd (or any other) sequence on the next step. Furthermore, a single sequence on time step $t$ can produce multiple sequences selected for step $t + 1$.

Since different beam search variations use different criteria for selecting top sequences, we need a generic algorithm that can fit any criterion. In our system, we implement this by allowing clients to reorder server-side attention cache after each step. Formally, a client can send a list of at most $k$ integers in range $[1, k]$, where i-th index specifies which previous attention cache should be used when generating $i$-th sequence of the next beam.

For instance, when given indices $[2, 2, 1, 3, 2]$, a server will use 2nd best sequence from step $t$ to produce the new 1st, 3rd and 5th best sequences. Previous 1st and 3rd best sequences go to 3rd and 4th places, respectively. Finally, previous 4th and 5th sequences are discarded. From a technical point of view, servers implement this reordering by reordering attention cache with the specified indices (`torch.gather` operation) immediately before performing an inference step.

Table 4: Zero-shot accuracy for BLOOM-176B and OPT-175B with 8-bit and 16-bit weights.

| Model | Bits | HellaSwag | LAMBADA | WinoGrande | Avg |
|-------|------|-----------|---------|------------|-----|
| BLOOM | 16 | 73.0 | 67.2 | 70.1 | 70.1 |
|       | 8  | 72.8 | 68.1 | 70.1 | 70.3 |
| OPT   | 16 | 78.5 | 74.7 | 72.6 | 75.3 |
|       | 8  | 78.5 | 74.6 | 71.7 | 74.9 |

Table 5: Generation throughput (tokens/s) for BLOOM-176B with 8-bit and 16-bit weights on $8\times$ A100 GPUs.

| Weights | Batch size | | |
|---------|------|------|------|
|         | 1 | 8 | 32 |
| 16-bit | 4.18 | 31.3 | 100.6 |
| 8-bit | 3.95 | 29.4 | 95.8 |

# D    Details of the server load balancing algorithms

**Measuring throughput.**    Before joining for the first time, each server measures its Internet connection throughput (in tokens/second, using one of public web APIs for doing that) and GPU throughput (in tokens/second, using a small benchmark running several forward passes). The minimum of these values becomes the overall server throughput, which is then cached for future runs.

**Initial block assignment.**    We assume that each server holds a segment of **consecutive** transformer blocks to minimize inference latency. Clients may request to perform a forward or backward pass for the whole segment of blocks or its subsegment, if necessary. Normally, each server loads as many blocks as it can fit in its GPU memory, unless a user limits the number of blocks to utilize the rest of memory for something else.

Before starting, each server calculates the values of $t_i$ – the total throughput of servers currently holding the $i$-th block or loading it (to start holding it in a few minutes). Then, to find the best segment of blocks to serve, the server looks for the most narrow bottleneck in the network. Formally, if the model has $L$ blocks and the server can hold $K$ of them in its GPU memory, we calculate:

$$start = \underset{i=1}{\overset{L-K+1}{\arg\min}} \quad \text{sorted}([t_i, \ t_{i+1}, \ \ldots, \ t_{i+K-1}]) \tag{1}$$

Here, $\arg\min$ compares the sorted arrays lexicographically and chooses the leftmost $start$ in case of multiple minimums.

This way, the next joining server would always cover a block with the smallest $t_i$. If there are multiple bottlenecks like this, the server will try to cover as many of them as possible (we choose to cover the minimums first because the overall throughput is the minimum of throughputs among model blocks). Among the remaining options, we choose a segment covering as many second minimums as possible, and so on.

**Quality of block assignment.**    While we are not aware of the exact polynomial-time solution for the problem of assigning the segments optimally, we have conducted computational experiments and found out that this greedy algorithm (running in polynomial time) usually finds an assignment with total throughput of 90-100% of the optimal one (found by trying out all possible assignments in exponential time), given that the values of throughput are realistic to our setup.

**Rebalancing.**    Since servers may leave at any time, each server also periodically checks if the current assignment is "good enough" compared to the throughput estimated by running the greedy solution for servers currently present in the network.

Formally, each server periodically looks for a segment of blocks that is more appropriate than the currently loaded blocks with respect to the $\arg\min$ rule (1). If it finds one, it simulates how the rest of the servers would behave if we replace the current blocks with the new ones (how other servers would change their blocks afterwards). If the eventual throughput is at least $p\%$ better, the server commits to the change and announces that it changes the blocks, then other servers do the rest of the changes (eventually increasing the total throughput).

We use $p = 20\%$ since it gives a reasonable trade-off between the swarm throughput and the frequency of block replacements in our experiments (see Appendix E). Specifically, a lower value of $p$ leads to block replacements happening too often, which negatively affects the inference latency since each block replacement resets attention caches for this block.

**Stability of the greedy algorithm.**    The rebalancing algorithm does not cause oscillations since a series of block replacements is executed only if it leads to eventually increasing throughput by at least $p\%$. Once a "good enough" throughput is achieved, servers do not change their blocks anymore (unless an essential number of servers join or leave). We verified this behavior computationally, simulating a network with thousands of servers with different throughputs.

To conclude, this greedy heuristic allows servers to quickly close the gaps if a substantial share (up to 100%) of servers holding certain blocks leave, but avoids excess block replacements otherwise.

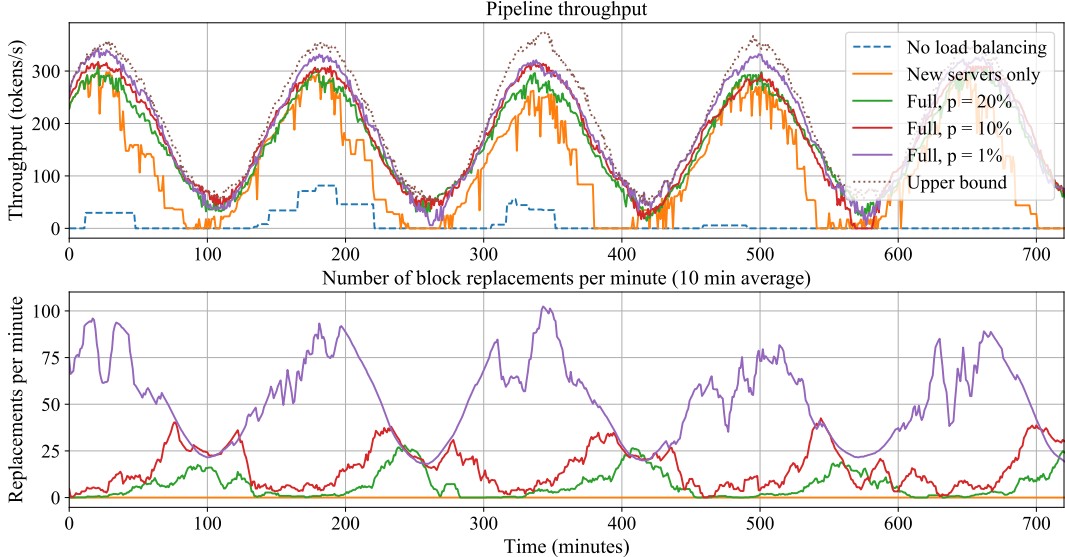

Figure 2: Behavior of the load balancing algorithms evaluated in Appendix E.

# E   Evaluation of the server load balancing algorithms

In this section, we measure the effectiveness of the load balancing algorithm used in our system. We run all experiments using a fleet of 206 virtual instances that simulate participants. To keep experiment costs manageable, we do not use GPUs for this evaluation, instead simulating uneven server throughput programmatically. For each server, we sample its throughput from the uniform distribution $t \sim \mathbb{U}[0, 100]$ tokens/second, then sample its memory size so it can hold $b \sim \mathbb{U}[1, 10]$ blocks (out of 70 blocks in total, as in BLOOM-176B).

Each server follows a certain availability schedule, i.e. turns on and shuts down at the same predefined time across all experiments. We assign these schedules such that the number of active servers follows a sine wave, simulating daily activity cycles. The schedule has approximately 100–110 active servers during peak activity and 15–25 servers at its lowest points. Note that each peak contains a different subset of 100–110 active servers out of 206 instances in total.

We evaluate the following approaches to load balancing:

1. **No load balancing** – a baseline system where servers load a random contiguous interval of model blocks.
2. **Balancing new servers only** – a simplified load balancing where servers choose the optimal blocks when joining the swarm (using the rule (1) from Appendix D) but never change them.
3. **Full load balancing** – the full algorithm, where every minute each server checks if they need to replace their blocks. We use the efficiency threshold $p$ (as described in Appendix D) to avoid excess block replacements.
4. **Upper bound** — the best-case throughput estimate that reassigns contiguous block segments to servers optimally every minute.

We report their behavior in Figure 2. The full load balancing maintains connectivity throughout the experiment and achieves throughput close to the upper bound (staying within the 10–15% range most of the time). Higher thresholds $p$ perform slightly worse during peak times but require only relatively infrequent block replacements, unlike the case with $p = 1\%$. Note that using the assignment leading to the upper bound is not possible in practice since it requires each server to load a different set of layers every minute, on top of solving the computationally expensive optimization problem.

Curiously, the baseline running load balancing for *new servers only* achieves reasonable throughput during periods where servers are actively joining. However, it quickly loses throughput when random servers leave, since this creates "bottlenecks" in the pipeline that require rebalancing of existing peers. Finally, the naive baseline with random layer assignment has zero throughput most of the time because it is unable to form a complete pipeline.

# F   Experiments with a wider range of failure rates

In this section, we follow the setup from Section 4.1 and provide additional evaluations for a wider range of failure rates (up to 5%) and sequence lengths (up to 2048 tokens). The results are shown in Figure 3. Unlike baselines, our algorithm provides reasonable performance *in all tested conditions*, especially for higher failure rates common for communicating over the Internet, using spot/preemptible instances or unreliable hardware).

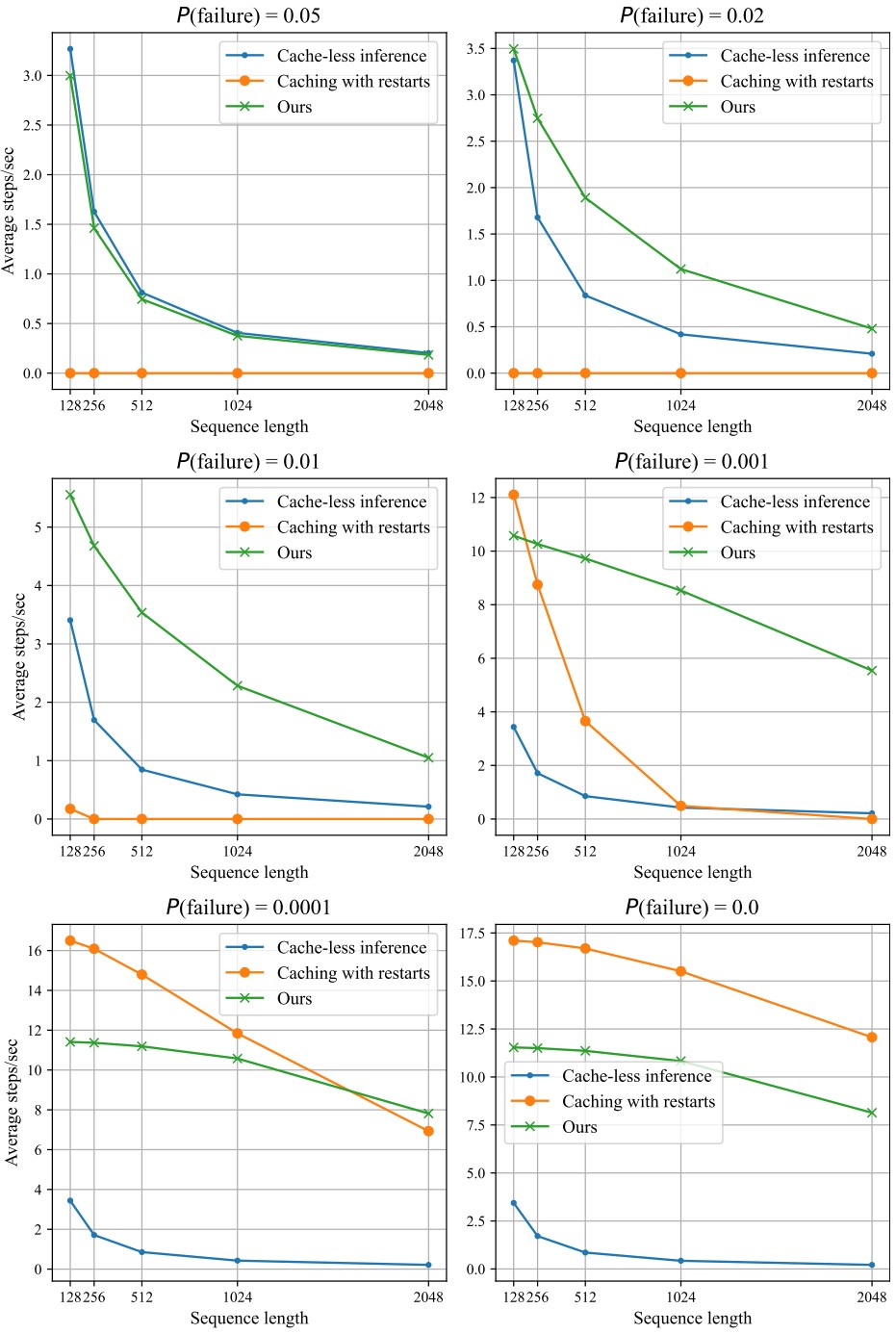

Figure 3: Sequential inference speed (steps/s) for BLOOM (7.1B) with varying failure rates. The setup is the same as in Section 4.1. A failure rate $p$ means that sending a set of activations to the next pipeline stage fails with probability $p$. Zero speed means that the baseline did not finish within 1 hour.

Table 6: Throughput (tokens/sec) of forward and backward passes for different tasks, batch sizes, prefix lengths.

| Mode | Batch size | Prompt length | Forward pass throughput | Backward pass throughput |
|---|---|---|---|---|
| Prompt tuning | 8 | 16 | 195.6 | 57.4 |
| | 8 | 4 | 213.2 | 60.8 |
| | 32 | 16 | 272.6 | 82.8 |
| | 32 | 4 | 293.1 | 84.7 |
| Prefix tuning (i.e., "deep" prompt tuning) | 8 | 16 | 111.0 | 42.0 |
| | 8 | 4 | 178.7 | 57.8 |
| | 32 | 16 | 164.1 | 64.4 |
| | 32 | 4 | 255.8 | 84.8 |

## G  Performance of training-time forward and backward passes

In this section, we evaluate throughput of training-time forward and backward passes and study factors that affect their performance. We will only consider BLOOM-176B and the "$3\times$ A100, 1 Gbit/s" setup from Section 4.2 and focus on finetuning-specific hyperparameters, since the influence of network bandwidth and latency has already been discussed in the main paper.

**Sequence classification.**  First, we consider fine-tuning the model on a binary classification task. We take BLOOM-176B, replace the logit layer with a trainable classification head (similar to `transformers.BloomForSequenceClassification`), and add trainable prompts before the input sequence, then train the model on batches of 128-token sequences. We try **(a)** both prompt tuning and prefix tuning (involving "deep" prompts), **(b)** two batch sizes (8 and 32), and **(c)** two prompt lengths (16 and 4). The client shares 8 CPU cores with one of the servers and does not use the GPU.

The results are provided in Table 6. The prefix tuning turns out to be slower, since it adds several times more trainable parameters. Increasing prompt length and decreasing batch size also make training slower. Notably, we observe that moving client-side computations to GPU does not visibly improve performance, since the client does not perform any heavy operations in this setup[11].

**Language modeling.**  Next, we consider fine-tuning the model on a causal language modeling task. We take BLOOM-176B, keep the logit layer, and add trainable prompts before the input sequence. We explore the same hyperparameters as with sequence classification.

We observe that the throughput of the GPU-enabled client is similar (within 10% difference) to the throughput in case of sequence classification, reported in Table 6. Indeed, the client performs only a small share of GPU computations in the forward and backward passes, and a particular model head and a loss function do not have decisive influence on the performance. However, performance of the CPU-only client turns out to be 5-10 times worse in this setup, since the client has to multiply the output embedding matrix to the hidden states of all tokens in the batch. This operation is too large to be efficiently computed on CPU[12].

## H  Limitations and broader impact

**Privacy.**  A key limitation of our approach is that servers hosting the first model blocks may use their inputs to recover client data. Thus, users working with *sensitive* data should limit their clients to only use trusted servers or, alternatively, set up their own isolated network using our software. For example, if multiple research labs or small companies have access to a specific private dataset

---

[11]In case of sequence classification, the heaviest operation the client does is multiplying $2 \times h$ and $h \times b$ matrices, where $h$ is the hidden dimension (14336 in BLOOM-176B) and $b$ is the batch size.

[12]In case of language modeling, the client has to multiply $d \times h$ and $h \times b$ matrices, where $d$ is the token vocabulary size (250880 in BLOOM-176B). This is $\approx 10^5$ times more FLOPS than used in case of sequence classification.

and want to process it with a large language model, they may set up an isolated distributed network hosting this model to get a better inference speed, compared to running the model independently.

In the future, this limitation may be addressed in future work using secure multi-party computing (Evans et al., 2018) or privacy-preserving hardware (NVIDIA, 2022).

**Motivating contributors.**    Since people using the client are not required to run a server, our system may experience an imbalance between supply (peers who dedicate GPUs to serve model layers) and demand (peers using the servers to perform inference or fine-tuning for their own needs).

One way to encourage users to serve model blocks would be to introduce a system of incentives: peers running servers would earn *reward points*, which can be spent on high-priority inference and fine-tuning or exchanged for other rewards. To implement this, we can run a few *validator peers* that periodically traverse all available servers and issue reward points to their owners.

**Security.**    We assume that servers in our system are run by many independent parties. In practice, some of them may turn out to be faulty and return incorrect outputs instead of the actual results of forward and backward passes. This may happen due to a malicious intent to influence other people's outputs or, when rewards are introduced (as described above), to earn a reward for serving layers without actually performing the calculations.

To address this issue, we can extend the validator peers, so that they periodically test servers with random requests of different types and ban them if they respond with incorrect outputs (possibly, revoking their rewards). The validator requests should be difficult to distinguish from requests of typical users, so that malicious servers cannot pretend to be honest to the validators but send wrong outputs to other peers. While this approach still leaves a chance of receiving wrong outputs, it allows to eventually expose and penalize the faulty servers.

Finally, clients may reduce the probability of getting faulty outputs by running their data through multiple disjoint chains of servers simultaneously and comparing the outputs against each other.

**Broader impact.**    This work introduces a general-purpose algorithm for decentralized inference and fine-tuning of large models, aiming to simplify access to the latest research in deep learning and provide an alternative way to efficiently run LLMs without high-end hardware. We do not envision any direct negative impacts from our research, since models that can be hosted with our system are already widely available and may be used via APIs, offloading, or other means.

