# OpenReview forum: "Distributed Inference and Fine-tuning of Large Language Models Over The Internet"
_NeurIPS.cc/2023/Conference — NeurIPS 2023 poster_

### Official Review · Reviewer_8b6R · 2023-07-04

**Soundness:** 3 good
**Presentation:** 3 good
**Contribution:** 3 good
**Rating:** 5
**Confidence:** 4

**Summary:**

The paper proposed cost-efficient inference and fine-tuning methods for LLMs on geodistributed devices in a consumer-grade network. The motivation is that, by pooling together idle compute resources of multiple research groups and volunteers, we could make LLM research and applications accessible to broader communities. Technically. this paper comes up with an algorithm to address two challenges: 1) how to conduct the computing reliably if any device can disconnect abruptly; 2) how to to partition LLMs between devices with uneven hardware. According to their simulations and real-world experiments, the proposed method can outperforms other approaches to running inference on consumer-grade hardware.

**Strengths:**

- The motivation of this paper is realistic. As we all know that playing with LLM is costly regarding the computing resources. Using the idle resources to do LLM inference and fine-tuning is socially and environmentally good.
- The idea of this paper is clear and practical.
- The paper applied multiple optimizations from different dimensions regarding the training/inference of LLM under low-resource, e.g., quantization both weights and activations between pipeline stages, efficient fine-tuning and so on. Although each of these methods are not really new, it is still inspiring to put them altogether and show that they work well.
- The experiments are reasonable, especially the real-world setup.


**Weaknesses:**

1. How to do fine-tuning under the proposed setting is not that clear, although the authors wrote one paragraph to explain the fine-tune part. Inference is clear and relatively simple, but how to recover from the failure during the training is not described and verified.
2. The communication cost from the tensor parallel is kind of missing. It seems that authors assume each server/client is able to hold a pipeline stage. However, in a more realistic scenario, each stage is further divided into multiple parts and one server/client could only hold a part of a stage. Then the intensive communication of tensor parallel may dominate the inference speed, because usually the tensor parallel only happen within one server, instead of distributed devices.
3. In Table 1, why the performance of Cache-less is irrelevant to the failure rate?


**Questions:**

Please refer to the Weaknesses

**Limitations:**

The potential limitations include privacy of data processed by outside peers, as well as broader impact of making LLMs more accessible.

---

> ### Author Rebuttal · Authors · 2023-08-10
>
> Thank you for reviewing the paper and leaving valuable feedback. We address the raised concerns below.
>
> > How to do fine-tuning under the proposed setting is not that clear, although the authors wrote one paragraph to explain the fine-tune part. Inference is clear and relatively simple, but how to recover from the failure during the training is not described and verified.
>
> Unlike inference, fine-tuning forward and backward passes process the entire batch at one go and **do not need to store past attention caches** between successive client requests. Thus, in case of a failure, we discard the incomplete forward/backward pass and just repeat the previous forward/backward pass request. This algorithm behaves similarly to the cache-less inference baseline in Table 1. We will extend the paper to clarify that in the next update.
>
> We also provide **fine-tuning experiments in Appendix F.1**, verifying our algorithm experimentally.
>
> > The communication cost from the tensor parallel is kind of missing. It seems that authors assume each server/client is able to hold a pipeline stage. However, in a more realistic scenario, each stage is further divided into multiple parts and one server/client could only hold a part of a stage. Then the intensive communication of tensor parallel may dominate the inference speed
>
> We run our experiments without tensor parallelism, since one transformer block of the largest open LLMs available at the moment (such as BLOOM-176B and OPT-175B) fits into 3 GiB of memory with 8-bit quantization. Thus, most consumer GPUs compatible with deep learning software **can hold a pipeline stage** with at least 1-2 transformer blocks.
>
> We do not use tensor parallelism over the Internet since its communication overhead is indeed too large. However, **our algorithm allows to use tensor parallelism across GPUs on the same machine** to speed up pipeline stages hosted on machines with multiple GPUs, since communication cost between machine's GPUs is much lower compared to the Internet. Our algorithm can treat such machines as a single "virtual GPU" with higher performance due to parallel computations. In this case, the system will take intra-host communication overheads into account while evaluating the machine's total compute throughput (later used for load balancing and fastest-inference routing).
>
> > In Table 1, why the performance of Cache-less is irrelevant to the failure rate?
>
> In case of failure, the cache-less baseline only needs to retry the last generation step and does not need to recover anything. For failure rate $p$, the number of retries for each step follows the geometric distribution with success probability $1 - p$ with the expected value of $1 / (1 - p)$.
>
> Thus, the entire process slows down by $1 / (1 - p)$ times, which turns out to be not significant for the setup from Table 1 with failure rates $p \le 10^{-2}$. However, cache-less inference is highly inefficient for longer sequences, which is not appropriate for most LLM use cases.

---

### Official Review · Reviewer_zN15 · 2023-07-04

**Soundness:** 3 good
**Presentation:** 3 good
**Contribution:** 3 good
**Rating:** 7
**Confidence:** 4

**Summary:**

The paper discuss about an important application problem of distributed inference for large language models. Given the size and inference requirement for large language models, and the constraint of hardware resources, the authors put forward utilizing idle GPUs in network to sped up the inference, providing detailed algorithm implementation and real world experimentation. Overall the work is solid and the presentation is good, and the application has important real world use cases. I have a few minor comments below.

**Strengths:**

1. real world problem of focus - the target problem to be resolved is important
2. solid experimentation - across continents experiment to demonstrate the distributed accelerator system's performance is amazing
3. detailed implementation - algorithm detail, result comparison, and analysis are solid

**Weaknesses:**

Overall I feel given the topic of distributed inference for LLM, the work is solid and clear. Some aspects to improve the work includes

1. server utilization discussion - the paper lacks coverage regarding the resource utilization of different GPU servers, given the distributed and heterogeneous computing setting
2. distributed accelerator infra requirement for LLMs - depending on different LLMs, there should be some basic requirement on the hardware. For example TPU is not covered, high-end CPUs are also not explored
3. expansion to LLM training - this is not the weakness of work, but would be great if the direction can be explored
4. consumer-grade network constraint - similar to 2, there are other real world constraints given the topic for the "consumer-grade network". To push the work beyond lab setting, network constraints like different firewalls, fault-tolerance mechanism considering failure rate beyond 1% (the upper bound of experimentation setting), etc

**Questions:**

see above comments

**Limitations:**

see above comments

---

> ### Author Rebuttal · Authors · 2023-08-10
>
> Thank you for reviewing the paper and leaving valuable feedback. We address the raised concerns below.
>
> > server utilization discussion - the paper lacks coverage regarding the resource utilization of different GPU servers, given the distributed and heterogeneous computing setting
>
> Resource utilization is indeed an important concern with two main aspects: memory and compute. The **memory utilization was above 90%** for all GPUs used in Section 4.2. Most of this memory is allocated for storing LLM parameters, the rest is used for activations and attention caches.
>
> In turn, the compute utilization rate depends on many factors including network bandwidth, client activity, and the exact workload. To that end, we evaluate several different server configurations with varying GPU and bandwidth: **(1)** RTX 3060 with 100 Mbit/s bandwidth, **(2)** RTX 3090 with 500 Mbit/s bandwidth and **(3)** RTX 3090 with 100 Mbit/s bandwidth. We assign each server to run forward passes for batches of 128 tokens sent by multiple concurrent clients (up to saturation), using the same workload as in Appendix F.1. We measure volatile GPU utilization (see [1]) averaged over 100 consecutive samples after a 1 minute warmup. We observe an average of **91% utilization for RTX 3060 and 100 Mbit/s, 94% utilization for RTX 3090 and 500 Mbit/s, and 66% utilization with 100 Mbit/s bandwidth.**
>
> [1] https://developer.download.nvidia.com/compute/DCGM/docs/nvidia-smi-367.38.pdf
>
> > distributed accelerator infra requirement for LLMs - depending on different LLMs, there should be some basic requirement on the hardware. For example TPU is not covered, high-end CPUs are also not explored
>
> **Server requirements.** The only GPU requirement for a server node is to have enough memory for one pipeline stage (= one transformer block). This is not an issue for most GPUs — only 3 GiB are needed for the largest open LLMs available (BLOOM-176B, OPT-175B), given that we use 8-bit quantization.
>
> Running on CPUs is possible but most CPUs are an order of magnitude slower. TPUs should work in principle, but are not supported by our software stack.
>
> **Client requirements.** The client node only computes input and output embeddings and does not require an accelerator, unless we perform fine-tuning with a computationally expensive loss function. The only requirement is to have enough memory for the embeddings (8 GiB are needed for BLOOM-176B embeddings in bfloat16).
>
> > expansion to LLM training
>
> Our system does support fine-tuning (see Section 3.4, experiments in Appendix F.1). As for pre-training from scratch, a similar setup was explored in other recent work [1, 2]. However, in contrast to our system, these methods are not designed to **(a)** run autoregressive inference with low latency and fault tolerance and **(b)** allow users to fine-tune the distributed model for multiple different tasks simultaneously.
>
> [1] Yuan, Binhang, et al. "Decentralized training of foundation models in heterogeneous environments." Advances in Neural Information Processing Systems 35 (2022): 25464-25477.
>
> [2] Ryabinin, Max, et al. "Swarm parallelism: Training large models can be surprisingly communication-efficient." arXiv preprint arXiv:2301.11913 (2023).
>
> > To push the work beyond lab setting, network constraints like different firewalls, fault-tolerance mechanism considering failure rate beyond 1% (the upper bound of experimentation setting), etc
>
> We explore other challenges, such as NATs, firewalls, and using heterogeneous hardware, in the "Real-world setup" experiments (L318-321). In particular, we show that **the servers are able to traverse NATs and firewalls** using the Circuit Relay protocol from libp2p, by opening a long-living connection to another directly available peer and asking it to become a relay.
>
> We also report plots showing the behavior of our algorithm **for failure rates 2% and 5%** in the general response PDF. We can see that it still has competitive performance in these cases. We will include these plots in the next paper revision.

---

### Official Review · Reviewer_2M8A · 2023-07-05

**Soundness:** 3 good
**Presentation:** 3 good
**Contribution:** 3 good
**Rating:** 6
**Confidence:** 5

**Summary:**

The objective of this study is to facilitate the operation of Large Language Models (LLMs) using commodity hardware over the internet. However, such hardware can often be characterized by high unreliability and latency issues in networks. To mitigate these challenges, the paper introduces a dual attention caches method that backs up intermediate results and supports device failure recovery. Additionally, the authors have developed a decentralized load-balancing algorithm to optimally assign transformer blocks to each server in a bid to maximize the system's overall throughput. The authors have provided a robust implementation of the proposed system and demonstrated impressive performance on the largest publicly available open-source LLM.

**Strengths:**

* Offloading parameters of LLM to remote devices instead of local local storage (e.g. SSD) make senses. Although the former could have higher bandwidth, the IO amount could be much higher than the later. Emprical results also support this analysis.
* The analysis on inference chanllenges is interesting. For example, the communication cost, past token storage. It helps the community understand the chanllenges of serving a large language model with billions of parameters.
* client/server caching, shorest path routing, and automatic load balancing improve robustness, effiiency and soundness of the system.

**Weaknesses:**

* I found that Table 2 is a bit vague to follow. Please eborate the metrics steps/sec and tokens/sec per user. For example, the difference between step and tokens and between clients and users.

**Questions:**

* For the comparsion between distributed inference and local offloading in table 2, does the local offloading also use the qunatized version of BLOOM? A clarification on this will help understand where the improvement of distributed inference comes from.

**Limitations:**

* Precise and explicit definitions of client and server are missing.
* Missing references on offloading inference compute across multiple devices [1,2, 3]. Discussing these would provide a more comprehensive context and enhance the depth of analysis.

[1] Kang, Yiping, et al. "Neurosurgeon: Collaborative intelligence between the cloud and mobile edge." ACM SIGARCH Computer Architecture News 45.1 (2017): 615-629.

[2] Matsubara, Yoshitomo, et al. "Head network distillation: Splitting distilled deep neural networks for resource-constrained edge computing systems." IEEE Access 8 (2020): 212177-212193.

[3] Dong, Xin, et al. "Splitnets: Designing neural architectures for efficient distributed computing on head-mounted systems." Proceedings of the IEEE/CVF Conference on Computer Vision and Pattern Recognition. 2022.

---

> ### Author Rebuttal · Authors · 2023-08-10
>
> Thank you for reviewing the paper and leaving valuable feedback. We address the raised concerns below.
>
> > I found that Table 2 is a bit vague to follow. Please eborate the metrics steps/sec and tokens/sec per user. For example, the difference between step and tokens and between clients and users.
>
> Table 2 reports the speed of autoregressive inference and parallel forward passes that **each client gets** on average. We assume that each user runs one client, so there is no difference between "clients" and "users" in this context.
>
> For inference, the speed is measured in generation steps per second each client can do (we use batch size 1, so each step generates 1 token), showing **generation latency**. For parallel forward, the speed is measured in tokens per second each client can process, showing the swarm's throughput during **batch processing and/or fine-tuning**.
>
> We will update the table caption and titles to clarify this in the next revision.
>
> > For the comparsion between distributed inference and local offloading in table 2, does the local offloading also use the qunatized version of BLOOM?
>
> Yes, we mention that in L267 after the paragraphs about quantization.
>
> > Precise and explicit definitions of client and server are missing.
>
> We provide short definitions in L150-152 and will expand them to the more explicit definitions provided below in the next revision.
>
> A **client** is a node operated by the user, which runs inference or fine-tuning jobs through the swarm of servers. A client only holds input and output embeddings (< 3% of model weights for BLOOM-176B) and delegates running transformer blocks (the most expensive computations) to remote servers.
>
> A **server** is a GPU-enabled node holding a set of consecutive transformer blocks and processing requests coming from client nodes.
>
> > Missing references on offloading inference compute across multiple devices [1, 2, 3].
>
> Thank you for making us aware of this work. We focused on optimizing distributed computation of common model architectures widely used in practice today, such as BLOOM/Falcon, OPT, and LLaMA. We think the work that you mentioned provides an interesting approach that highlights how new architectures can be used in distributed computing. We will add discussion of these papers to the related work section.

---

### Official Review · Reviewer_DRjo · 2023-07-10

**Soundness:** 3 good
**Presentation:** 3 good
**Contribution:** 3 good
**Rating:** 6
**Confidence:** 3

**Summary:**

This paper presents a system designed for decentralized inference and fine-tuning of large language models over distributed hardware, which allows users to efficiently run LLMs without requiring high-end hardware. The system leverages pipeline-based model parallelism, distributing model layers across nodes. Additionally, this work proposes fault-tolerant inference algorithms and load-balancing protocols to enable dynamic deployment of models. During the evaluation, experiments were conducted using the BLOOM-176B model, which demonstrated a 10x improvement in performance compared to the offloading method.

**Strengths:**

The writing of this paper is fluent, with clear and logical reasoning, and the proposed solution is well-aligned with the requirements；
The design is consistent with current trends and can effectively address the problem, making it highly practical；
This paper presents a comprehensive system, with reasonable comparisons made against upper bounds and benchmarks.



**Weaknesses:**

The innovative aspects of the paper are not sufficiently elaborated upon. For example, I would like to know if there are any outstanding advantages compared to the latest research such as DeepSpeed, aside from differences in application scenarios. Has there been any comparison of similar metrics?

The results are somewhat unsatisfactory. Caching with restarts appears to be quite competitive, and the algorithm proposed in the paper performs better only under high failure rates. However, for short sequences, Cache-less inference performs better under high failure rate conditions (which may be due to some inherent communication issues in system). It would be helpful to compare several lengths or plot curves to more clearly show the trend and find the optimal point.

Although the focus of the research is on the inference process, it would be best to systematically elaborate on the experimental results and methods related to fine-tuning for the sake of experimental completeness.

A minor suggestion is to cite Algorithms 2/3 more specifically in the text to make the writing clearer.

I am not particularly familiar with distributed learning, so please forgive me if I make any inaccuracies in my statements.

**Questions:**

The paper mentions that autoregressive LLM inference cannot be performed with a single pass through the model, leading to a higher system complexity. Do the proposed algorithms still have advantages for models that are not autoregressive? Was this the primary design point of the system, or is it also applicable to other models?

During the evaluation of offloading, how many GPUs were used for benchmarking? Is there a strict basis for the best-case scenario?

**Limitations:**

(1)To better evaluate the system, it would be beneficial to extend the experiments to include more models.
(2)Data privacy is a concern, as multiple clients may contribute to data misuse. It would be helpful to propose some solutions to address this issue.
(3)Furthermore, the innovative aspects of the paper could be further clarified.

---

> ### Author Rebuttal · Authors · 2023-08-10
>
> Thank you for reviewing the paper and leaving valuable feedback. We address the raised concerns below.
>
> **Weaknesses**
>
> > The innovative aspects of the paper are not sufficiently elaborated upon. For example, I would like to know if there are any outstanding advantages compared to the latest research such as DeepSpeed, aside from differences in application scenarios. Has there been any comparison of similar metrics?
>
> Our paper focuses on an algorithm for running LLMs using a swarm of globally distributed GPUs connected over the Internet, addressing challenges of unreliable network and hardware. Standard pipeline parallelism methods like DeepSpeed can't work in this setup since they are designed for high-speed networked GPU clusters. Thus, we offer **a novel, cheap way to efficiently run LLMs for people without high-end hardware** — they can collaborate with other researchers and join their GPUs over the Internet to host the full model. This was **not possible with previously existing methods** and software.
>
> We do not expect our algorithm to beat existing methods on a local, reliable GPU cluster, since this is not a setup it was designed for. Still, **we provide comparison with local pipeline parallelism that uses DeepSpeed** in Table 2 (see "Local PP (NVLink)", details in L309-311) to demonstrate the overheads that the proposed geo-distributed setup has compared to a local GPU cluster (expensive hardware that is not available to everyone).
>
> > Caching with restarts appears to be quite competitive, and the algorithm proposed in the paper performs better only under high failure rates. [...] It would be helpful to compare several lengths or plot curves to more clearly show the trend and find the optimal point.
>
> The paper proposes the general-purpose system that has to successfully operate on both short and long sequences for various failure rates. The baselines are indeed competitive at some operating points but are highly impractical at others.
>
> We agree that the plots would be useful and provide them in the general response **PDF**. Unlike the baselines, our algorithm provides **reasonable performance for all tested conditions**, especially for higher failure rates (common for communicating over the Internet, using spot/preemptible instances or unreliable hardware). We will include the plots in the next paper revision.
>
> > it would be best to systematically elaborate on the experimental results and methods related to fine-tuning for the sake of experimental completeness.
>
> Please see **fine-tuning experiments in Appendix F.1**. Unlike inference, fine-tuning processes the entire batch at one go and does not need to store past attention caches between successive client requests. Thus, in case of a failure, we discard the incomplete forward/backward pass and just repeat the previous forward/backward pass request. We will elaborate on this in the main text in the next revision.
>
> **Questions**
>
> > Do the proposed algorithms still have advantages for models that are not autoregressive?
>
> Our system is applicable to arbitrary models, not only autoregressive ones. In our opinion, using it may be beneficial when: **(1)** the model does not fit into most customer GPUs (i.e., you can't load it locally), **(2)** single-batch computation time is much faster when the entire model is present in the GPU memory, compared to copying model weights to GPU on demand (i.e., you can't use offloading efficiently).
>
> These points are especially relevant for autoregressive LLMs due to their huge size, and the extra need to maintain past attention caches makes the problem even more pronounced and challenging.
>
> > During the evaluation of offloading, how many GPUs were used for benchmarking? Is there a strict basis for the best-case scenario?
>
> Note that the bottleneck for offloading-based LLM inference is the GPU bus throughput, not the GPU performance (or the number of GPUs). For inference, the **best-case speed estimate** for offloading considers the GPU bus throughput only (for best existing hardware) and **assumes infinite GPU performance — and still turns out to be 5-10x slower** than our system. We provide step-by-step calculations in Appendix B.
>
> The offloading experiments use 1x A100 (see Table 2 in Section 4.2) or 1x 3090 (see Table 6, Appendix F.2) and fully confirm our theoretical analysis. They show that **(1)** the **real offloading performance is even smaller** than the estimated best-case performance due to compute costs and other overheads (0.0495 < 0.18 steps/sec) and **(2)** the offloading performance **does not depend much on the GPU model**, having similar performance for both A100 and 3090 (0.0495 vs 0.0427 tokens/sec).
>
> **Limitations**
>
> > it would be beneficial to extend the experiments to include more models
>
> We agree and provide experiments with **Llama 2 (70B)** in the general response. We will include them in the next paper revision.
>
> > Data privacy is a concern, as multiple clients may contribute to data misuse. It would be helpful to propose some solutions to address this issue.
>
> We acknowledge this limitation in the paper (L346) and discuss potential solutions, such as using privacy-preserving computation methods or setting up a private swarm between trusted parties, in **Appendix G.**

---

> > ### Comment · Reviewer_DRjo · 2023-08-11
> >
> > Firstly, I appreciate the authors' efforts in addressing my concerns with detailed explanations. Regarding the suggestion about applicability across different models, it was a more general remark, and I commend the authors for promptly conducting the experiments. The timely addition of the final improvement effect graph and the willingness to acknowledge the limitations in the work are also notable. I have accordingly adjusted my evaluation score.

---

### Author Rebuttal · Authors · 2023-08-10

We thank all reviewers for taking the time to study our paper and leave valuable feedback. We are glad that the reviewers appreciated the motivation behind our work (*8b6R, zN15*), our analysis of LLM inference challenges (*2M8A*), the soundness of the proposed system for geo-distributed inference (*2M8A, 8b6R*), and our experimental work (*zN15, 8b6R*).

**Experiments with Llama 2 (70B).** Following the request of reviewer *DRjo*, we provide experiments with Llama 2 (70B) below. We run our system on 3 machines with one T4 GPU (16 GB memory) each and compare its performance to offloading running on each of these machines independently. The model is quantized to the NF4 format [1] in both cases. All other setup details are the same as in Section 4.2 and Table 2.

| GPUs | Parallel clients | Bandwidth, RTT     | Single-batch inference |         | Parallel forward |                 |
|------|------------------|--------------------|------------------------|---------|------------------|-----------------|
|      |                  |                    | steps/s for each client |        | tokens/s for each client |         |
|      |                  |                    | **128 tokens** | **2048 tokens** | **Batch 1x128** | **Batch 64x128** |
|      | 1                | 1 Gbit/s, < 5 ms   | 2.29           | 2.02            | 45.4            | 155.1            |
|      | 1                | 100 Mbit/s, < 5 ms | 2.29           | 2.01            | 37.5            | 140.2            |
| 3x T4 | 1                | 100 Mbit/s, 100 ms | 1.57           | 1.44            | 23.7            | 128.7            |
|      | 3                | 1 Gbit/s, < 5 ms   | 2.02           | 1.74            | 21.2            | 124.2            |
|      | -                | Offloading         | 0.139          | 0.139           | 18              | 139.9            |

We can see that our system still **beats offloading by more than 10x for inference**, even when all clients run inference simultaneously. Our system is also faster at fine-tuning in case of smaller batches or good network bandwidth. Thus, **all conclusions from Section 4.2 hold** for this setup.

[1] Dettmers, Tim, et al. "QLoRA: Efficient finetuning of quantized LLMs." arXiv preprint arXiv:2305.14314 (2023).

**Experiments with more failure rates.** Following the feedback from reviewers *DRjo* and *zN15*, we attach a **PDF with plots** that report our system's performance for a wider range of failure rates (including > 1%) and sequence lengths. Unlike baselines, our algorithm provides **reasonable performance in all tested conditions**, especially for higher failure rates (common for communicating over the Internet, using spot/preemptible instances or unreliable hardware).

**Limitations.** Finally, reviewers *8b6R* and *DRjo* noted data privacy as a limitation of our work. We acknowledge it in the paper (L346) and discuss potential solutions, such as setting up a private swarm between trusted parties or using privacy-preserving computation methods, in **Appendix G**.

---

### Decision · Program_Chairs · 2023-09-21

**Decision:**

Accept (poster)

**Comment:**

The reviewers and AC discussed this paper and read the authors’ response carefully. All the reviewers unanimously agreed to accept this paper.

The authors are encouraged to address the concerns from the reviewers in the camera ready, such as:

1.	Adding communication costs

2.	Expansion to LLM training

3.	Improve the presentation of Table 2.